# Exploration of Alternative Splicing (AS) Events in MDV-Infected Chicken Spleens

**DOI:** 10.3390/genes12121857

**Published:** 2021-11-23

**Authors:** Lulu Wang, Gang Zheng, Yiming Yuan, Ziyi Wang, Changjun Liu, Hao Zhang, Ling Lian

**Affiliations:** 1Department of Animal Genetics and Breeding, College of Animal Science and Technology, China Agricultural University, Beijing 100193, China; LULU131522@163.com (L.W.); jiujiu0820@163.com (G.Z.); yuanyimingnice@163.com (Y.Y.); wangzydL6@163.com (Z.W.); zhanghao827@163.com (H.Z.); 2Division of Avian Infectious Diseases, Harbin Veterinary Research Institute, Chinese Academy of Agricultural Sciences, Harbin 150001, China; liuchangjun@caas.cn

**Keywords:** marek’s disease, alternative splice, RNA-Seq, regulatory network

## Abstract

Marek’s disease (MD) was an immunosuppression disease induced by Marek’s disease virus (MDV). MD caused huge economic loss to the global poultry industry, but it also provided an ideal model for studying diseases induced by the oncogenic virus. Alternative splicing (AS) simultaneously produced different isoform transcripts, which are involved in various diseases and individual development. To investigate AS events in MD, RNA-Seq was performed in tumorous spleens (TS), spleens from the survivors (SS) without any lesion after MDV infection, and non-infected chicken spleens (NS). In this study, 32,703 and 25,217 AS events were identified in TS and SS groups with NS group as the control group, and 1198, 1204, and 348 differently expressed (DE) AS events (*p*-value < 0.05 and FDR < 0.05) were identified in TS vs. NS, TS vs. SS, SS vs. NS, respectively. Additionally, Function enrichment analysis showed that ubiquitin-mediated proteolysis, p53 signaling pathway, and phosphatidylinositol signaling system were significantly enriched (*p*-value < 0.05). Small structural variations including SNP and indel were analyzed based on RNA-Seq data, and it showed that the TS group possessed more variants on the splice site region than those in SS and NS groups, which might cause more AS events in the TS group. Combined with previous circRNA data, we found that 287 genes could produce both circular and linear RNAs, which suggested these genes were more active in MD lymphoma transformation. This study has expanded the understanding of the MDV infection process and provided new insights for further analysis of resistance/susceptibility mechanisms.

## 1. Introduction

Marek’s disease (MD), induced by Marek’s disease virus (MDV), caused huge economic losses to the global poultry industry [1]. There were four infection stages in MD development, including the early cytolytic phase, latency phase, late cytolytic and immunosuppressive phase, and visceral tumors formation phase [2]. Currently, the most effective way to control MD was injecting MDV vaccines to 1-day-old chicks. However, vaccines can just prevent neuropathy, oncogenic disease, and immunosuppression. They could not prevent MDV transmission or infection [3]. Additionally, the use of vaccines promoted the shift of the virus to very virulent, and vaccinated individuals also could be infected by virulent strain [4]. As a lymphoma disease induced by an oncogenic virus, MD was regarded as an ideal model for studying diseases induced by the oncogene virus. MDV showed some similar features with human herpesvirus 6 (HHV-6) [5] and Epstein Barr virus (EBV) [6], such as they all can integrate into the host genome [7,8,9] and were related to lymphoproliferative cancers [10,11]. Moreover, MDV was similar to human herpesvirus 1 (HSV-1) and human herpesvirus 3 (VZV) from the genetic point of view [9]. Hence, many studies focused on elaborating the mechanism of MDV infection from the perspective of genetics to provide a basis for the development of vaccines and disease treatment induced by the oncogenic virus.

In the human genome, only 2% of genes could be translated into proteins, while the remaining 98% were noncoding RNAs without coding potential [12]. Alternative precursor-mRNA (pre-mRNA) splicing (AS) contributed to producing various noncoding transcripts and proteins [13,14]. Noncoding transcripts included short microRNAs, piwi-interacting RNAs, small nuclear RNAs, circular RNAs, and long noncoding RNAs [12]. Different protein isoforms generated from alternative splicing possessed different biological properties such as protein interaction, subcellular localization, or catalytic ability [15]. Back-splicing product circRNAs also had various biological functions such as miRNA sponge [16], interacting with RNA binding protein (RBP) [17], regulating parental genes [18]. Studies reported that alternative splicing was associated with various diseases [13,19,20,21]. Moreover, alternative splicing resulted in function changes of chromatin-modifying enzymes [22,23], which affected chromatin modification to regulate gene expression from an epigenetic perspective [24,25]. It reported that meq bound to host genomic chromatin may contribute to MDV-mediated T-cell transformation and lymphomagenesis [26]. Jin et al. [27] analyzed the relationship between the alternative exon 7 splice variant of the BF gene and MD resistance. They found that tumor incidence and mortality were highly related to the alternative exon 7 of the BF gene. Jarosinski and Schat [28] found that there were multiple splice variants of meq, RLORF5a, and RLORF4 to exons II and III of vIL-8, and these alternatively spliced transcripts were the time-dependent expression. Kaya et al. [29] validated alternatively spliced transcripts of seven genes in line 6 (MD resistant) and seven (MD susceptible) using RNA-Seq data, including ITGB2, SGPL1, COMMD5, MOCS2, CCBL2, ATAD1, and CHTF18.

Indel (insertion-deletion length polymorphism) and SNP (single nucleotide polymorphism), the most abundant DNA polymorphisms in eukaryotic genomes [30], were associated with individual development and various diseases. A study showed that a CR1 SNP was associated with higher rates of medium-term disease progression [31]. Stone [32] found that five noncoding SNPs in the KLK6 region were strongly related to the aggressiveness of prostate cancer, which can serve as biomarkers for this disease. The frequency of the 12 and 23 bp indel polymorphisms in the PRNP in cattle were associated with the risk of bovine spongiform encephalopathy [33]. The study also showed that an indel polymorphism in 3′UTR of SPRN was significantly associated with scrapie positivity in the central nervous system. It may regulate scrapie susceptibility by miRNA-mediated post-transcriptional mechanism [34]. Studies reported that genomic variants were also related to MD resistance/susceptance. Li et al. [35] detected two SNPs that were associated with host resistance to MD. SNP located in GH6 was also associated with MD assistance [36,37]. Variants in the splice site region may be attributed to producing new transcripts by affecting splice sites or splice motifs. Functional SNP formed from NCF4 splice variants was associated with mastitis susceptibility in dairy cows, and NCF4 expression was also affected by alternative splicing [38]. However, there is no systematic study that analyzed the alternative splicing and variants in MD development based on the transcriptome level.

In this study, we aimed to comprehensively analyze the alternative splicing (AS) events and genetic variants in MDV-infected and non-infected spleens to identify some active candidate genes involved in MD tumor transformation, which provided new ideas both for chicken tumor disease and human cancer caused by an oncogenic virus.

## 2. Materials and Methods

### 2.1. Biological Samples

To identify AS events in chicken spleens during the MDV-induced transformation and development stage, we downloaded RNA-Seq data (GSE124133) of 17 samples from the Gene Expression Omnibus (GEO).

The specific information of experimental samples was described in the previous study [39]. Briefly, 150 1-d-old specific-pathogen-free White Leghorn (BWEL) chicks were divided into 2 groups. One group (*N* = 100) was infected with intraperitoneally with 2000 plaque-forming units (PFUs) of the MDV-GA, and the other group was mock-infected with the same volume of diluent (0.2 mL) as a control group (*N* = 50). Two groups were housed in different isolators. The tumor group is determined based on clinical symptoms and visceral tumor conditions, and samples of individuals in the control group are collected at the same time. The trial period lasted 56 days post infection, and all remaining birds without clinical phenotype were regarded as survivors (SS), and their spleens were collected. Seven tumor spleens (TS) in the tumor group, five normal spleens (NS) in a control group, and five spleens in the survivor group were sampled. The specific collection time points and the sex of samples are shown in Appendix A. All tissues were preserved in an RNA fixer at 4 °C; overnight and transferred to −80 °C for further study.

### 2.2. Alternative Splicing Analysis

To identify AS events of MDV-infected spleens, Illumina RNA-Seq data (GSE124133) was downloaded from the Gene Expression Omnibus (GEO).

The replicate multivariate analysis of transcript splicing (rMATS) v4.0.1 [40] was used to detect and analyze alternative splicing events in all samples. Briefly, the index of the reference genome (version: Gallus_gallus-6.0; GCA_000002315.5) was built using HISAT2, and paired-end clean reads were aligned to the reference genome using HISAT2 with default arguments [41]. SAMTOOLS [42] was used to convert aligned results to BAM format files. Then, BAM files in each group were submitted to rMATS to identify alternative precursor-mRNA (pre-mRNA) splicing (AS) with default arguments. Five AS events types, including skipped exon (SE), alternative 5′ splice site (A5SS), alternative 3′ splice site (A3SS), mutually exclusive exons (MXE), and retained intron (RI), could be detected. *P*-value, which was calculated by the likelihood-ratio test, represented the difference between the two groups of samples at the IncLevel (Inclusion Level). The FDR value was calculated by correcting the *p*-value using the Benjamini–Hochberg algorithm. AS events with the threshold of *p*-value < 0.05 and FDR < 0.05 were differential AS events.

### 2.3. Function Enrichment Analysis

Gene lists were submitted to the database for annotation, visualization, and integrated discovery (DAVID 6.8) for gene ontology (GO) enrichment and KEGG pathway analysis. GO terms and KEGG pathway with *p*-value < 0.05 were significantly enriched. String v11.0 was used to construct a protein-protein interaction network with default parameters. The hub genes in the network were identified by maximal clique centrality (MCC) method of cytoHubba module in Cytoscape software.

### 2.4. Analysis of SNP and Indel

Briefly, BWA software was used to perform alignments with the genome and generated SAM files. SAM files were converted and sorted to BAM files via SAMTOOLS. Genome Analysis Toolkit [43]) was used to generate genomic variants files (GVCF) and SNP and indel information. SNP and indel were filtered by GATK with DP < 5.0, QUAL < 30.0, QD < 2.0, FS > 200.0, SOR > 10.0, MQRankSum < −12.5, ReadPosRankSum < −20.0. VCFTOOLS [44] was also used to filter again with the threshold --max-missing 0.9 --maf 0.01. SNPEFF [45] was used to annotate SNP and indel.

### 2.5. Validation of AS Events

Four AS events were randomly selected to validate the accuracy of the AS results. PCR was performed with cDNA of spleen tissues from TS, NS, and SS groups as the templates, and specific primers were shown in Appendix A. PCR products were detected by 5% agarose gel electrophoresis and then sequenced.

## 3. Results

### 3.1. Identification of Alternative Splicing Events

In the present study, we identified 32,703 and 25,217 AS events in TS and SS groups using rMATS with the NS group as the control group (Figure 1A, Appendix A). In the TS group, 27,172 SE events, 4300 MXE events, 671 RI events, 237 A5SS events, and 323 A3SS events were identified, which corresponded to 8747, 2458, 592, 217, and 296 genes. In the SS group, 21,366 SE events, 2700 MXE events, 626 RI events, 220 A5SS events, and 305 A3SS events were identified, which corresponded to 8025, 1698, 555, 202, and 273 genes (Figure 1B). Compared to other AS events types, SE events were the most frequent AS types, and one gene could produce three forms of SE events on average.

### 3.2. Validation and Expression Analysis of AS Events

Four AS events were validated by PCR, agarose gel electrophoresis, and Sanger sequencing (Figure 2). PCR results showed that SLA, NRP2, SVIL, and RUNX2 both had two products, and the length of the product met with the length of inclusion transcript and skipping transcript. Exon-skipped sequences were also detected via Sanger sequencing. Combining with sequence counts, there were more inclusion junction counts than skipping junction counts of SLA, SVIL. However, inclusion junction reads were less than skipping junction reads of NRP2 (Table 1), which was consistent with the agarose gel electrophoresis result.

### 3.3. Analysis of SNP and Indel

Totally, 2,384,056, 2,563,255, and 2,387,444 SNPs and 543,143, 582,539, and 569,509 indels were detected in NS, SS, and TS groups (Figure 3A). Many variations were annotated in the transcripts (Figure 3B,D). Additionally, there were 105,350, 112,441, and 129,024 indels and 123,009, 131,338, and 148,219 SNPs located in splice region (including splice site acceptor, splice site donor, splice site region) (Figure 3C,E) in NS, SS, and TS groups.

### 3.4. Identification of Differentially Alternative Splicing (DE AS) Event and Corresponded Genes

DE AS events were detected by rMATs with the threshold of *p*-value < 0.05 and FDR < 0.05. A total of 2750 DE AS events were produced from 1231 genes (Figure 4A, Appendix A). Out of which, 527 genes generated one DE AS event, while other genes could produce more than two DE AS events (Figure 4B). In TS vs. NS, 1198 DE AS events were identified, including 999 SE events, 20 RI events, 156 MXE events, 16 A5SS events, and 7 A3SS events (Figure 4A), which corresponded to 840, 20, 138, 15, and 7 genes (Figure 4C). In TS vs. SS, 1015 SE events, 20 RI events, 142 MXE events, 14 A5SS events, and 13 A3SS events were identified as DE AS events (Figure 4A), which corresponded to 854, 20, 129, 14, and 13 genes (Figure 4C). In SS vs. NS, 348 DE AS events out of 27,753 AS events were detected, including 306 SE events, 10 RI events, 20 MXE events, 8 A5SS events, and 4 A3SS events (Figure 4A), which corresponded to 282, 10, 19, 8, and 4 genes (Figure 4C).

### 3.5. Function Enrichment Analysis of DE AS Corresponding Genes

To illustrate the role of DE AS corresponding genes on the MD development, GO enrichment and KEGG pathway analysis were produced (Appendix A). Significantly enriched GO terms and pathways were shown in Figure 5 with the threshold of *p* < 0.05. In TS vs. NS and TS vs. SS, several immune-related GO terms were significantly (*p* < 0.05) enriched, including ATP binding, GTPase activator activity, protein serine/threonine kinase activity, and protein autophosphorylation. In SS vs. NS, 15 GO terms were significantly (*p* < 0.05) enriched, such as intracellular signal transduction, extracellular matrix organization, positive regulation of cytokinesis, mitotic spindle assembly, and cell morphogenesis. KEGG pathway analysis showed several tumorigenesis-related pathways were significantly enriched (*p* < 0.05), including ubiquitin-mediated proteolysis, regulation of autophagy, NOD-like receptor signaling pathway, p53 signaling pathway, and phosphatidylinositol signaling system.

### 3.6. Identification of Genes between DE AS Corresponding Genes and CircRNA Parental Genes

Alternative splicing can produce circRNAs, which play important regulatory roles in tumor development, so we analyzed the overlap between alternative splicing transcripts and circRNA. To identify active genes in the MD tumor transformation process, we checked the genes producing both circular RNA and linear alternative splicing transcripts combined with previous circular RNA data. There were 287 overlapped genes when compared DE AS events corresponding genes with circRNA parental genes (Figure 6A). These 287 genes could produce 501 circular RNAs, and most of which were generated from the exon region (Figure 6B, Appendix A). Furthermore, 152 out of 287 genes produced 1 circular RNA, while RABGAP1L and DENND1A could generate 14 and 12 circular RNAs, respectively (Figure 6C).

Additionally, a PPI network of 287 genes was constructed. The top five nodes ranked by MCC were marked (Figure 6D). They were TRIP12, UBE2D1, ANAPC10, PJA2, and FBXL4. The TRIP12 gene could directly interact with 20 genes in the network and harbored 3 DE AS events (Figure 6E). Integrating splicing factors information, we found that there were some splicing factors among these 287 genes, including PTBP2 and MBNL1.

## 4. Discussion

Marek’s disease (MD), an immunosuppressive disease, induced by MDV with many clinical signs, including neurological symptoms, chronic wasting, lymphoma development in the viscera and musculature, and blindness [46]. Many viral genes in MDV genome have been identified that may involve in MD development, such as, meq [47], viral telomerase RNA (vTR) [48], viral interleukin-8 (vIL-8) [49], phosphoprotein 38 (pp38) [50,51], and MDV miRNAs [52]. For host genome, some genes involved in MD pathogenies were identified, including MHC [27,53], gga-miR-155 [54], gga-miR-26a and NEK6 [55], IGFBP7 and TNFRSF8 [39]. Several studies also illustrated the mechanism of susceptive or resistance from the epigenomic level [56,57,58]. Alternative splicing transcripts were also detected in MD pathogenies [27,29].

In this study, we comprehensively analyzed alternative splicing information in spleen tissues with MDV infection. With non-infected spleen as the control group, 32,703 and 25,217 AS events were identified in TS and SS. Only 24,356 genes existed in the chicken genome (version: Gallus_gallus-6.0; GCA_000002315.5), which means some genes produced more than one transcript. Variants in the splice region may result in alternative splicing [59]. Results showed that the TS group had more AS events than that in SS and NS groups, which may be resulted from its abundant variants in the splice region. The previous study showed that a function SNP of MRPL43 could regulate the splicing of MRPL43, which resulted in the decrease in major isoform and affected the balance between phosphorylation and glycolysis [38]. The 465 C>T SNP of NQO1 at the 5′-splice site of intron-4 promoted alternative splicing and decreased protein expression of NQO1 [60]. An SNP,18174 A>G, in the NCF4 gene induced 48 bp retained fragment in intron 9, which regulated NCF4 expression [38].

In this study, there were a few DE ASs (348) detected between SS and NS, which illustrated that compared with the TS group, gene expression patterns of individuals in the SS group were closer to that in the NS group. Differentially alternative splicing events corresponded genes MSH4 and MYCBP2 produced 16 and 13 AS events, respectively. MSH4, a meiosis-specific MutS homolog, was involved in DNA mismatch repair [61]. Zhou et al. [62] reported that MSH4 in the members of the nasopharyngeal carcinoma (NPC) clustering families upregulated over 1038 times than that in NPC patients. Tang et al. [63] reported that a novel homozygous stop-gain mutation of MSH4 was related to non-obstructive azoospermia. Human MutS homolog MSH4 played an important role in the maintenance of chromosomal stability by physically interacting with von Hippel-Lindau tumor suppressor-binding protein 1 [64]. MYCBP2, a member of the c-myc oncogene family [65], was overexpressed in T-cell lymphoma with a poor prognosis [66]. MYCBP2 mRNA expression was negatively correlated with survival in colorectal cancer (CRC) [65]. A study reported that low MYCBP2 expression was associated with high-risk acute lymphoblastic leukemia (ALL) [67]. In this study, MYCBP2 and MSH4 possessed up to 10 AS events, which indicated that they may play important roles in MD pathogenesis.

Function enrichment analysis of DE AS corresponding genes showed that terms related to disease pathogenesis were also enriched, including protein serine/threonine phosphatase activity [68,69], protein autophosphorylation [70], and protein tyrosine phosphatases (PTPs) that were essential regulators of signal transduction and may provide valuable therapeutics for human disease treatment [71]. We found that GO terms enriched in TS vs. NS were similar to that in TS vs. SS, but there was a few DE AS and enriched GO terms in NS vs. SS. KEGG pathway showed that pathways involved in various diseases were also enriched, such as ubiquitin-mediated proteolysis was related to Parkinson’s disease [72], clear cell renal cell carcinomas (RCCs) [73], regulation of autophagy, NOD-like receptor signaling pathway mediated the initial innate immune response to cellular injury and stress [74], p53 signaling pathway [75,76]. The previous study showed that MDV activated PI3K/Akt signaling through the interaction of its meq protein with the regulatory p85 subunit of PI3K to promote viral replication [77]. In this study, these disease-related GO terms and pathways were enriched, which means that there were many genes that regulated MD pathogenesis via participating in various immunity processes.

We speculated that genes simultaneously produced linear spliced transcripts, and back spliced transcripts were more active and played crucial important roles in MD pathogenesis via multiple mechanisms. In this study, 501 circRNAs, generated from 287 DE AS corresponding genes, were identified. RABGAP1L, the tyrosine-kinase target in signaling transduction [78], could predictively generate 14 circRNAs. RABGAP1L was involved in various human diseases, such as AF4-RABGAP1L, and RABGSP1L-MLL fusion transcripts were identified in t(4;11) leukemia patients [79], and methylation level was different in 18 babies born with congenital ZIKV microcephaly from 20 controls [80]. DENND1A, a member of the connection family, served as a component of clathrin-mediated endocytosis machinery [81], which could produce 12 predicted circular isoforms in our study. DENND1A was a candidate gene of polycystic ovary syndrome (PCOS) [82,83], which processed two variants, V1 (1009 AA) and V2 (559 AA), V2 mRNA and protein were overexpressed in PCOS theca cells compared to normal theca cells [84]. Additionally, DENND1A was also related to gastric cancer [84], endometrial carcinoma [85].

To find hub genes that can simultaneously produce variable splicing and circular RNA among these genes, a PPI network was constructed, and the MCC method in cytohubba was used to find the top five nodes of the network. Results showed that top 1 node TRIP12 could interact with other 20 genes, including UBE2D1, ANAPC10, PJA2, FBXL4. TRIP12, an E3 ubiquitin ligase, played a key role in cell cycle progression and chromosome stability [86], DNA damage response [87], and cell differentiation [88]. Additionally, Gao et al. [89] analyzed aberrant splicing events in acute myeloid leukemia (AML), and the results showed that exon3-skipping isoform of TRIP12 increased significantly after treatment. One of its interacted genes, UBE2D1, a member of ubiquitin-conjugating enzyme E2D family, related to the Aurora kinase A (AURKA), which was an enhancer of Wnt and Ras-MAPK signaling in colorectal cancer (CRC) [90]. Differential mRNA expression of UBE2D1 was related to patient survival [91]. The study showed that UBE2D1 significantly upregulated in lung adenocarcinoma (LUAD) and lung squamous cell carcinoma (LUSC), and it served as an independent prognostic indicator of unfavorable OS and RFS in LUAD [92]. Studies showed that UBE2D1 mediated the ubiquitination and degradation of p53, a tumor suppressor protein [93,94]. Zhou et al. [95] also reported that UBE2D1 significantly upregulated in HCC tissues and precancerous lesions, and it promoted HCC growth in vitro and in vivo by downregulating the p53 in the ubiquitination-dependent pathway. FBXL4, a nuclear gene, is involved in various diseases [96]. In this study, these disease-related genes simultaneously produced linear mRNA transcripts and circular RNA transcripts, which indicated that they regulated MD pathogenesis via the various pathway, including producing different proteins [84,97], serving as miRNA sponge [16], interacting with RNA binding protein (RBP) [17], regulating parental genes [98]. A large number of alternative splicing in tumor samples implies the abnormality of tumor samples. Some active genes played crucial roles in tumor transformation, and their functions needed to be further verified, which also provided a reference for human diseases induced by the virus.

## 5. Conclusions

In our study, 32,703 and 25,217 AS events were, respectively, identified in TS and SS groups. DE AS events analysis showed that many diseases related pathways were significantly enriched. Compared with the TS group, gene expression patterns of individuals in the SS group were closer to that in the NS group, suggesting that maintenance of, or return to, homeostasis of gene activity in survivors’ spleens. More AS evens and more variants on the splice site region were predicted in the TS group, which suggested the variants were involved in disease progression by influencing gene splicing to form different transcripts. This study has expanded our understanding of the process of MDV infection and provided new insights for further analysis of resistance/susceptibility mechanisms from the molecular biology level.

## Figures and Tables

**Figure 1 genes-12-01857-f001:**
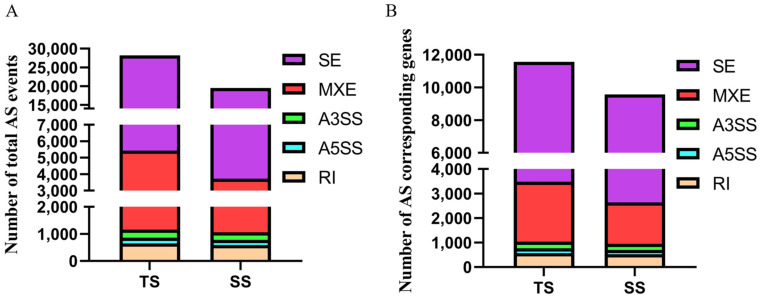
(**A**) The number of AS events in each group. (**B**) The number of corresponded genes of total AS evens in each group.

**Figure 2 genes-12-01857-f002:**
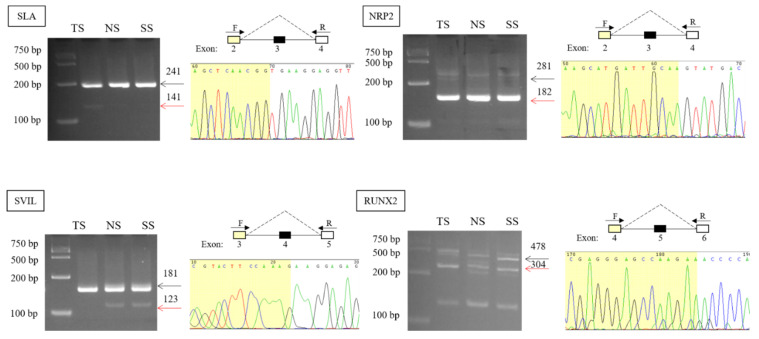
PCR amplification was performed TS, NS, and SS with primers designed for both sides of the skipped exons. The black arrow represents the product of amplification from the original transcript, and the red arrow represents the product of amplification after exon skipping. Sanger sequencing showed the sequence after exon skipping.

**Figure 3 genes-12-01857-f003:**
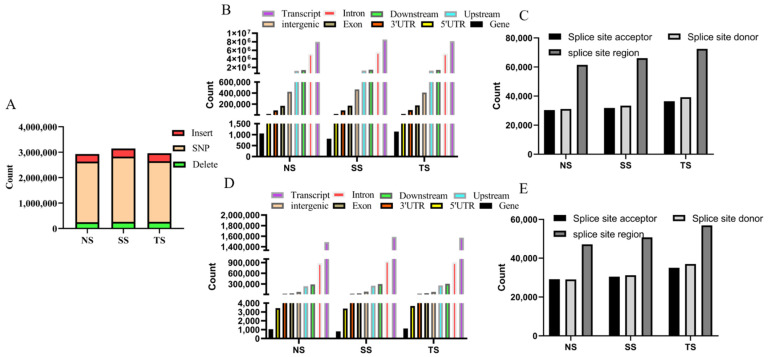
(**A**) The number of SNP and indel in TS, NS, SS groups. (**B**) Annotations of SNP on the genome. (**C**) Annotations of SNP on splice site acceptor, splice site donor, and splice site region. (**D**) Annotations of indel on the genome. (**E**) Annotations of indel on splice site acceptor, splice site donor, and splice site region.

**Figure 4 genes-12-01857-f004:**
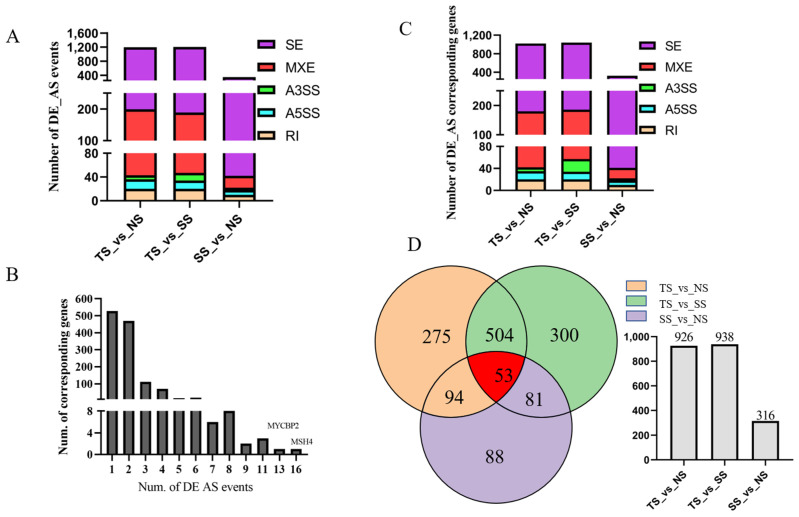
(**A**) Total DE AS events in TS_vs_NS, TS_vs_SS, SS_vs_NS. (**B**). Statistics on the number of alternative splicing events produced by parental genes. MYCBP2 possessed 13 predicted DE AS events, and MSH4 harbored 16 predicted DE AS events. (**C**). DE AS events corresponding genes in TS_vs_NS, TS_vs_SS, SS_vs_NS. (**D**) Venn diagram of DE AS events.

**Figure 5 genes-12-01857-f005:**
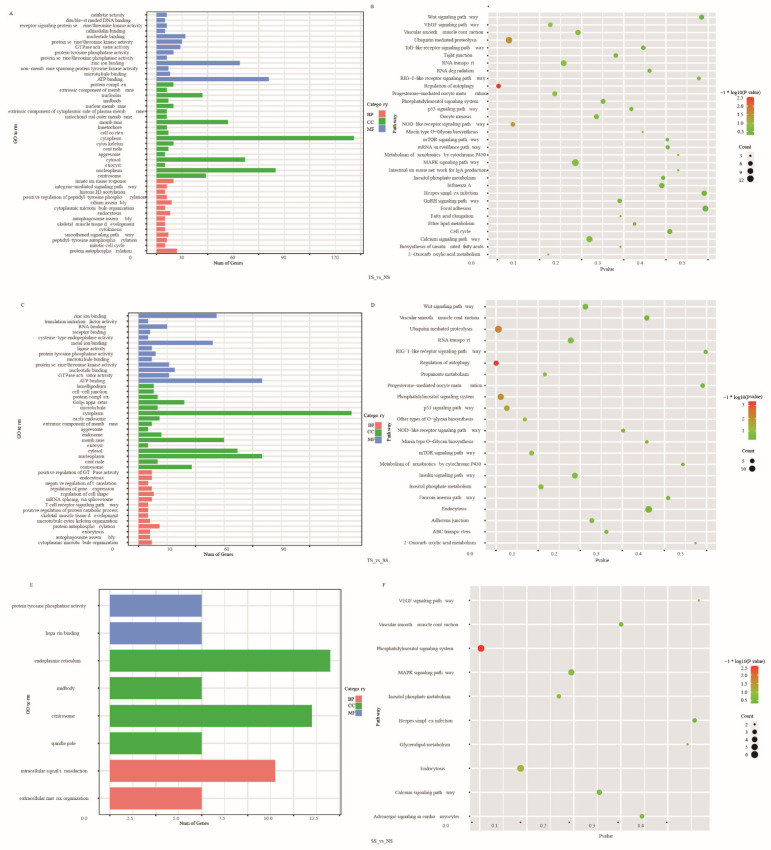
(**A**,**B**) Significantly enriched GO terms and KEGG pathway analysis of DE AS corresponded genes TS_vs_NS. (**C**,**D**) Significantly enriched GO terms and KEGG pathway analysis of DE AS corresponded genes TS_vs_SS. (**E**,**F**) Significantly enriched GO terms and KEGG pathway analysis of DE AS corresponded genes SS_vs_NS.

**Figure 6 genes-12-01857-f006:**
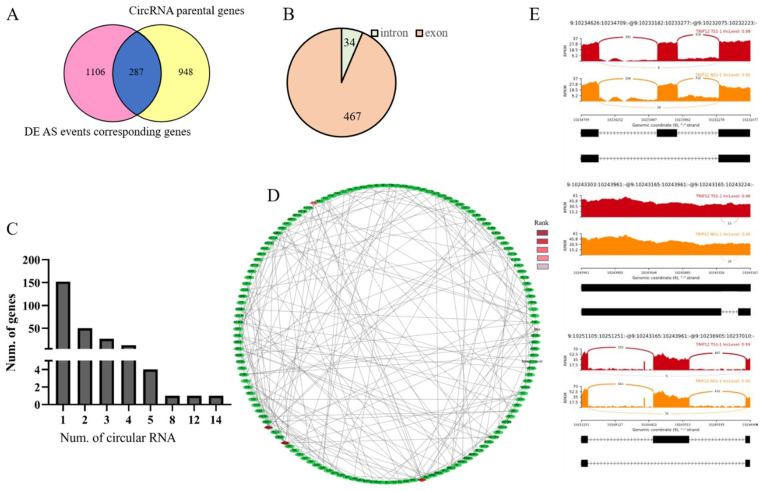
(**A**) A total of 287 overlapped genes in DE AS corresponded genes and circRNA parental genes. (**B**) The proportion of exon and intron circRNAs of 501 circRNAs generated from 287 overlapped genes. (**C**) Number of circRNAs that generated from these 287 genes. (**D**) PPI network of 287 genes. Top 5 nodes were identified by the MCC method in the cytohubba model in Cytoscape. (**E**) DE SE event of TRIP12 in TS vs. NS and TS vs. SS.

**Table 1 genes-12-01857-t001:** Inclusion junction counts (IJC) and skipping junction counts (SJC) of AS events in TS, NS, and SS.

Gene_Name	IJC_TS	SJC_TS	IJC_NS	SJC_NS	IJC_SS	SJC_SS
**SLA**	1730, 701, 912, 599, 833, 980, 1362	55, 43, 83, 19, 16, 28, 22	1379, 1025, 940, 1399, 1600	23, 9, 8, 17, 14	10, 229, 697, 271, 717, 700	14, 11, 68, 22, 6
**NRP2**	0, 58, 79, 22, 26, 15, 8	126, 272, 326, 146, 102, 57, 128	4, 1, 7, 0, 11	127, 129, 113, 204, 140	17, 11, 8, 20, 14	172, 182, 148, 184, 156
**SVIL**	658, 398, 738, 318, 133, 188, 482	20, 4, 4, 13, 18, 14, 36	327, 271, 220, 366, 296	43, 53, 60, 39, 33	220, 228, 289, 371, 427	29, 24, 28, 42, 34
**RUNX2**	45, 130, 38, 33, 53, 63, 66, 29	6, 47, 24, 3, 4, 6, 1	4, 7, 3, 5, 5	3, 0, 1, 1, 1	11, 13, 4, 7, 1	5, 0, 0, 2, 0

Note: The number that was separated by commas represented the counts in different samples were detected within the group.

## Data Availability

All data of this study are included in this published article and its Appendix A. RNA-Seq data are available at NCBI Gene Expression Omnibus (GEO, GSE124133).

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
