# Peer review of "Exploration of Alternative Splicing (AS) Events in MDV-Infected Chicken Spleens"

_genes, 2021, doi:10.3390/genes12121857_

Round 1
Reviewer 1 Report
Explain why the experimental period was scheduled to be 56 days, knowing the mechanism of neoplastic tumor formation in Marek's disease virus infection?
Explain why the experimental group of birds vaccinated against Marek's disease and infected with GA strain was not planned in the experiment?
Author Response
Q1: Explain why the experimental period was scheduled to be 56 days, knowing the mechanism of neoplastic tumor formation in Marek's disease virus infection?
A1: MD pathogenesis has four phases in the susceptible birds; An early cytolytic infection is started at 2-7dpi, a latent phase is initiated at around 7-10dpi with the MDV genome persisting in the host cells, a late cytolytic phase causes inflammation and transformation of latently infected lymphocytes into tumor cells and is triggered between 14-21dpi, a proliferative phase around 28 dpi is characterized by formation of visceral tumours that originate from CD4+ T cells lymphoma [1, 2]. Most of the current researches have focused on 5,10,14,21 days after infection [1, 3, 4]. Most researches regarding MD resistance were focused on infection phases ahead of 21dpi, while we were more interested in late transformation phase when malignant tumors burst which were serious damage to the host. The challenge trial was done in Harbin Veterinary Research Institute of Chinese Academy of Agricultural Sciences. Briefly, 150 chicks were divided into two groups. One hundred chicks were infected with MDV, and the other 50 were mock infected. During the trial, we observed the chickens’ clinical signs 2–3 times daily and severely morbid birds were considered as susceptible chickens and were euthanized. Tumors in multiple tissues could been observed in these chickens. The last morbid bird was observed and sampled at 50dpi. From then on there were no sick chicken found. After consulting with our co-author, Dr. Changjun Liu who is an expert in MD research field. He suggested 56dpi were long enough for identifying resistance chickens. So, we decided to euthanize all 21 remaining birds. In all these 21 chickens, there were 7 chickens showed MD symptoms including organs’ enlargement or tumors in tissues, the other 14 chickens had no lesion. Additionally, according to previous study, Marek's disease virus (MDV) infection in the eye was studied chronologically after inoculating 1-day-old chickens with a very virulent MDV strain, Md5. The ocular lesions could be classified as early lesions (6–11 dpi) and late lesions (26 and 56 dpi), based upon the location and severity of the lesions [5].
Q2: Explain why the experimental group of birds vaccinated against Marek's disease and infected with GA strain was not planned in the experiment?
A2: Thank you for your advice. In some MD studies, to investigate protective efficacy of MD vaccine, the authors did experiments that infecting vaccinated birds with MDV. For instance, Ralapanawe, et al. [6] studied the kinetics of Rispens CVI988 (Rispens) and two MDV strains of different virulence in 236 commercial ISA Brown chickens vaccinated with Rispens at hatch and challenged with vMDV isolate MPF57 or vvMDV isolate FT158 on day five. Results showed that Rispens vaccination significantly reduced challenge MDV viral load in a sample-dependant manner with evidence of a differentially greater inhibitory effect on the less virulent MDV.
The purpose of our research is to analyze the MD resistance or susceptible mechanism of host. If the post-immunization challenge was done, we cannot distinguish the effect of the vaccine or the disease resistance from the host itself.
References
- 1. Yu, Y., Luo, J., Mitra, A., Chang, S., Tian, F., Zhang, H.M., Yuan, P., Zhou, H.J., and Song J.Z. Temporal Transcriptome Changes Induced by MDV in Marek’s Disease-Resistant and -Susceptible Inbred Chickens. BMC Genomics.2011;21, 501.
- Boodhoo, N., Gurung, A., Sharif, S., and Behboudi, S. Marek's disease in chickens: a review with focus on immunology. Vet Res.2016;47, 119.
- Sun, G.R., Zhou, L.Y., Zhang, Y.P., Zhang, F., Yu, Z.H., Pan, Q., Gao, L., Li, K., Wang, Y.Q., Cui, H.Y., Qi, X.L., Gao, Y.L., Wang, X.M., and Liu, C.J. Differential expression of type I interferon mRNA and protein levels induced by virulent Marek's disease virus infection in chickens. Vet Immunol Immunopathol.2019;212, 15-22.
- Bavananthasivam, J., Alizadeh, M., Astill, J., Alqazlan, N., Matsuyama-Kato, A., Shojadoost, B., Taha-Abdelaziz, K., and Sharif, S. Effects of administration of probiotic lactobacilli on immunity conferred by the herpesvirus of turkeys vaccine against challenge with a very virulent Marek's disease virus in chickens. Vaccine.2021;39, 2424-2433.
- Pandiri, A., Cortes, A., Lee, L., and Gimeno, I. Marek's Disease Virus Infection in the Eye: Chronological Study of the Lesions, Virus Replication, and Vaccine-Induced Protection. Avian Diseases.2008;52, 572-580.
- Ralapanawe, S., Walkden-Brown, S.W., Islam, A.F., and Renz, K.G. Effects of Rispens CVI988 vaccination followed by challenge with Marek's disease viruses of differing virulence on the replication kinetics and shedding of the vaccine and challenge viruses. Vet Microbiol.2016;183, 21-29.
Reviewer 2 Report
The authors had successfully investigated alternative splicing (AS) events in MD by comparing the RNA sequences prepared from tumorous spleens (TS), spleens from the survivors (SS) without any lesion after MDV infection, and non-infected chicken spleens (NS). Some specific differently expressed AS among three groups were found and the corresponded genes were analyzed by function enrichment analysis (gene ontology (GO) enrichment and KEGG pathway analysis). The ubiquitin-mediated proteolysis, p53 signaling pathway, and phosphatidylinositol signaling system were significantly enriched (p-value < 0.05). The results showed a comprehensive investigation of AS events during MDV infection and provided more information about the mechanisms of resistance/susceptibility to the MDV in hosts from molecular biology level.
Suggestions:
- Line 21, “combined with circRNA “ changes to “Combined with circRNA “.
- Line 50, “various noncoding and proteins” might need to be revised.
- Line 173, Panel A and Panel B shall be exchanged.
- Line 198, “ 611 genes generated” In Figue 4C, I cannot figure out where the 611 from.
- Line 232-233, “play important roles in MD important roles” might need to be revised.
- Line 235, “178 genes produced” In Figue 6C, I cannot figure out where the 178 from.
- For all the figures, the font size is too small. Even enlarging the figures, the fonts were still hard to read.
- For all the figures, the explanation in the legend is not clear for reader to follow.
Author Response
Q1: Line 21, “combined with circRNA “changes to “Combined with circRNA “.
A1: Thank you for your advice. We have modified this description in line 23.
Q2: Line 50, “various noncoding and proteins” might need to be revised.
A2: “various noncoding and proteins” has been replaced by “various noncoding transcripts and proteins”. Please see lines 50-51.
Q3: Line 173, Panel A and Panel B shall be exchanged.
A3: Thank you for your suggestion. We have exchanged the position of these two panels.
Q4: Line 198, “611 genes generated” In Figue 4C, I cannot figure out where the 611 from.
A4: Sorry for our wrong description on this point. We checked the data again and found the data in figures were right, but the number in the main text were wrong. We have modified it in line 191.
Q5: Line 232-233, “play important roles in MD important roles” might need to be revised.
A5: we have corrected the description of this sentence and changed it to “play an important role in the transformation of MD tumor”
Q6: Line 235, “178 genes produced” In Figue 6C, I cannot figure out where the 178 from.
A6: Thank you very much. We have re-checked the data and it was an error. It should be 152 and we have revised it. Please see lines 231-232.
Q7: For all the figures, the font size is too small. Even enlarging the figures, the fonts were still hard to read.
A7: According to your suggestion, we have modified the font size in all figures.
Q8: For all the figures, the explanation in the legend is not clear for reader to follow.
A8: According to your suggestion, we have described the chart information in details.